# The relationship between Braden Skin Score (BSS) and the risk of acute respiratory distress syndrome in elderly sepsis patients: An analysis based on the MIMIC-IV database

Yingqi Xiao☯, Yi Cao☯, Jianwen Guo, Liufang Shu, Jingcheng Xu, Zhiyong Li, Yongchun Li [iD]*

Department of Infectious Disease, The Sixth Affiliated Hospital, School of Medicine, South China University of Technology, Foshan, China

☯ These authors contributed equally to this work.
* liyongchun2025@163.com

## Abstract

### Objective

The Braden Skin Score (BSS) is a validated tool for assessing pressure injury risk. This study aimed to investigate whether BSS can also predict the risk of developing Acute Respiratory Distress Syndrome (ARDS) in elderly sepsis patients in the intensive care unit (ICU).

### Background

ARDS is a common and serious complication in ICU patients with sepsis, associated with prolonged hospitalization, metabolic disorders, and increased mortality. Elderly sepsis patients are particularly vulnerable to pressure injuries due to prolonged bed rest. Nevertheless, the relationship between BSS and the risk of developing ARDS in this population has not been extensively studied.

### Methods

Elderly sepsis patients from the MIMIC-IV database were chosen and partitioned into quartiles of BSS values. Although the incidence of ARDS was the primary endpoint, hospital mortality in the ARDS subgroup was the secondary endpoint. The correlation between low BSS and risk of ARDS and in-hospital mortality in elderly sepsis patients was assessed using logistic regression models.

**Data availability statement:** All relevant data are within the manuscript and its Supporting information files.

**Funding:** The author(s) received no specific funding for this work.

## Results

Multivariate logistic regression analysis revealed that a lower BSS at ICU admission was significantly associated with an increased risk of ARDS and higher in-hospital mortality among elderly sepsis patients.

## Conclusion

This study demonstrates that a low BSS at ICU admission is associated with a greater risk of ARDS and higher in-hospital mortality in elderly sepsis patients.

## Introduction

Sepsis is one of the most common causes of death in ICU patients, particularly among the elderly [1]. Elderly patients are more susceptible to sepsis due to reduced physiological reserves and the development of multi-organ dysfunction. The high morbidity and mortality rates associated with sepsis are often attributable to its common and serious complication, acute respiratory distress syndrome (ARDS) [2]. Some factors linked to the development of ARDS include underlying diseases, the severity of the infection, and the body's immune response [3]. While existing research on ARDS has primarily focused on its etiology, pathophysiology, clinical manifestations, diagnosis, and treatment [4], a significant gap remains in prevention strategies for specific high-risk populations, such as elderly sepsis patients. Therefore, there is a need to identify practical predictive tools that can identify elderly sepsis patients at high risk of ARDS, enabling the development of personalized preventive measures to improve patient outcomes.

The Braden Score, which has recently been hypothesized as a surrogate measure of overall patient health and prognosis, is commonly used to assess the risk of pressure ulcers [5–7]. Several factors of the Braden Score, including mobility [8], sensory perception, and moisture, have recently been associated with the risk of other complications. This scoring system demonstrates excellent predictive capability for pressure injury risk in clinical settings, with lower scores indicating higher risk. Recent studies have demonstrated that the Braden Skin Score (BSS) can effectively predict the incidence of acute kidney injury in patients with acute coronary syndrome, overall mortality in acute myocardial infarction, mortality and post-discharge outcomes in hospitalized elderly patients, recovery following pancreatic resection, and outcomes in ICU patients with ischemic stroke [9–12].

Furthermore, studies have shown that ARDS is associated with various biomarkers and clinical scoring systems, such as the Lung Injury Prediction Score and coagulation dysfunction scores [13,14]. These scoring systems can help clinicians identify high-risk patients to some extent. However, the role of the Braden Score, as a simple and practical assessment tool, in predicting ARDS risk has not been fully explored. Therefore, based on the MIMIC-IV database [15], this study aims to investigate the relationship between the Braden Score and the risk of ARDS in elderly sepsis patients. The MIMIC-IV database is a large, publicly available ICU database

containing comprehensive clinical and laboratory data, which provides robust support for this study. Through an in-depth analysis of this database, we seek to elucidate the potential association between the Braden Skin Score and ARDS risk in elderly sepsis patients, ultimately offering clinicians a simple tool for early identification of high-risk patients and implementation of timely interventions to improve outcomes.

## Methods

### Data source

This study adopted a retrospective observational design. The cohort was derived from the publicly available clinical database, Medical Information Mart for Intensive Care IV (MIMIC-IV) (https://mimic.mit.edu). To comply with relevant regulations, the author, Yingqi Xiao, obtained the Collaborative Institutional Training Initiative (CITI) license and the necessary certification to use the MIMIC-IV database.

### Exposure variables

Data collection used Structured Query Language (SQL) and PostgreSQL (version 14.2) to extract patients' baseline characteristics. These characteristics included demographic data (age, sex, body mass index (BMI)), vital signs (heart rate (HR), systolic blood pressure (SBP), and diastolic blood pressure (DBP)), laboratory test results (white blood cell count (WBC), hemoglobin, platelets, serum albumin, serum creatinine (Scr), ALT, AST, and comorbidity information obtained from the MIMIC-IV database. We selected the exposure factors according to established practices in previous studies [16,17].

For missing values of the exposure variable, imputation methods were adopted (referring to a widely recognized convention in epidemiological and clinical research: covariates with a missing rate exceeding 25% were directly excluded to reduce potential bias that might arise from direct input of missing values; imputation methods were applied to covariates with a missing rate of less than 25%. After screening the variable data, no covariates with a missing rate exceeding 25% were identified) [18].

Chronic obstructive pulmonary disease (COPD), hypertension, atrial fibrillation (AF), and diabetes were defined using the International Classification of Diseases, Tenth Revision (ICD-10) and ICD-9 codes. The follow-up period began on the date of patient enrollment and ended when the endpoint of interest occurred.

### Outcome variables

The primary endpoint was the incidence of ARDS in elderly sepsis patients. According to the reference https://doi.org/10.1001/jama.2016.0287, Sepsis was defined as meeting the Sepsis-3.0 criteria [19], specifically a Sequential Organ Failure Assessment (SOFA) score ≥ 2 points and infection or suspected infection. ARDS was defined according to the Berlin criteria [20] for patients aged 18 years or older diagnosed with ARDS. The Berlin definition is as follows: (1) acute onset of respiratory symptoms; (2) bilateral chest imaging opacities; (3) arterial oxygen partial pressure (PaO2) to fractional inspired oxygen (FiO2) ratio < 300 mmHg, with positive end-expiratory pressure (PEEP) ≥ 5 cm H2O; (4) absence of heart failure.

The secondary endpoint included in-hospital mortality in the ARDS subgroup.

### Statistical analysis

**Association between BSS score and outcome variables.** Continuous variables were described using mean (standard deviation) or median (interquartile range), and group comparisons were made using rank-sum tests or t-tests based on the nature of the data. Researchers expressed categorical variables as frequencies and percentages (%) and performed group comparisons using Fisher's exact or Pearson's chi-square test. Quantitative data were analyzed using

t-tests or rank-sum tests, and categorical data were analyzed using chi-square tests to compare differences in exposure factors between groups. The preliminary analysis used Multivariate logistic regression to determine the association between BSS and outcome variables. Two-sided tests tested All descriptive analyses for significance at a significance level of $P < 0.05$.

All data analyses were performed using R.4.4.1 (http://www.R-project.org). The sample size was based on available data, with no pre-study sample size calculation.

## Results

### Demographic characteristics

This study focused on the MIMIC-IV study population. In cases of multiple hospital admissions, only the initial hospitalization was considered. To ensure data integrity, patients were excluded if their ICU stay was less than 48 hours, if BSS score data were missing within 24 hours of ICU admission, if ARDS diagnostic information was incomplete, or if follow-up data were missing. Ultimately, a final study cohort of 16,565 elderly sepsis patients was established (Fig 1), with raw data provided in S1 Table.

In the preliminary analysis of the MIMIC-IV elderly sepsis population, there were 6,793 participants in the ARDS group and 9,772 participants in the non-ARDS group. The ARDS group was younger than the non-ARDS group, with a statistically significant difference ([74.0 (66.9–81.8)] vs. [74.6 (67.4–82.4)], $p < 0.001$). There were no significant differences in gender distribution or BMI between the two groups. Regarding vital signs, the ARDS group had higher heart rates and diastolic blood pressure than the non-ARDS group, with statistically significant differences but no difference in systolic blood pressure. Laboratory tests showed that the ARDS group had higher white blood cell counts, hemoglobin, and platelets than the non-ARDS group, and disease severity scores (SOFA, APS III, etc.) were also higher in the ARDS group. During hospitalization, the mortality rate in the ARDS group was significantly higher than in the non-ARDS group (Table 1).

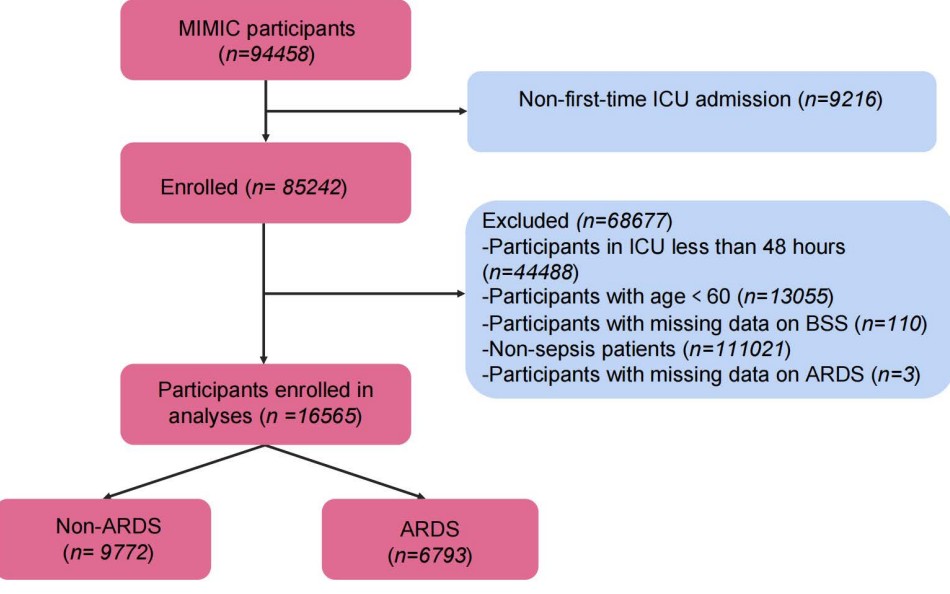

**Fig 1. Data screening flowchart.**

**Table 1. Characteristics of participants enrolled in study.**

| Characteristic | Non-ARDS (N = 9772) | ARDS (N = 6793) | P-value |
|---|---|---|---|
| Age (y) | 74.6 (67.4-82.4) | 74.0 (66.9-81.8) | 0.002 |
| Female sex | 4346 (44.5) | 2994 (44.1) | 0.611 |
| BMI (kg/m2) | 27.9 (24.5-32.1) | 27.9 (24.2-32.6) | 0.889 |
| Heart rate (beats/minute) | 83.2 (74.1-94.4) | 85.2 (74.3-97.7) | <0.001 |
| Systolic pressure (mmHg) | 112.7 (104.3-124.0) | 112.5 (104.6-123.7) | 0.816 |
| Diastolic pressure (mmHg) | 58.5 (52.9-64.9) | 59.6 (53.9-66.2) | <0.001 |
| WBC (K/uL) | 13.5 (9.70-18.30) | 14.1 (10.1-19.5) | <0.001 |
| Hemoglobin (g/dL) | 10.9 (9.6-12.3) | 11.1 (9.5-12.7) | <0.001 |
| Platelet (K/uL) | 203.0 (149.0-276.0) | 213.0 (152.0-288.0) | <0.001 |
| Albumin (g/dL) | 3.1 (2. 7-3.5) | 3.0 (2.6-3.4) | 0.018 0.018 |
| Serum creatinine (mg/dL) | 1.2 (0.9-2.0) | 1.4 (1.0-2.3) | <0.001 |
| ALT (U/L) | 28.3 (17.0-68.0) | 30.0 (18.0-69.0) | 0.025 |
| AST (U/L) | 45.0 (26.0-103.1) | 47.0 (28.0-111.0) | <0.001 |
| Diabetes | 779 (8.0) | 854 (12.6) | 0.094 |
| Hypertension | 7378 (75.5) | 5151 (75.8) | 0.63 |
| Atrial fibrillation | 4246 (43.5) | 3057 (45.0) | 0.092 |
| COPD | 779 (8.0) | 854 (12.6) | 0.048 |
| SOFA | 3.0 (2.0-4.0) | 3.0 (2.0-5.0) | <0.001 |
| APS III | 47.0 (36.0-61.0) | 54.0 (42.0-70.0) | <0.001 |
| SAPS II | 41.0 (35.0-50.0) | 46.0 (38.0-55.0) | <0.001 |
| OASIS | 4.0 (29.0-39.0) | 38.0 (33.0-44.0) | <0.001 |
| Hospital Death | 1317 (13.5) | 2256 (33.2) | <0.001 |

Note. Categorical data is displayed as n (%). Non-normal distribution data is displayed as median(Q1-Q3). χ2 analysis is used to test significance between groups for categorical data. Kruskal Wallis rank sum test is used to test significance between groups for non-normal distribution data. BMI = body mass index; WBC = white blood cell; ALT = alanine aminotransferase; AST = aspartate aminotransferase; SOFA = sequential organ failure assessment; APSIII = acute physiology score III; SAPSII = simplifed acute physiological score II; OASIS = oxford acute severity of illness score; COPD = chronic obstructive pulmonary disease.

## Association between BSS score and ARDS risk

In the overall population, the BSS score in the ARDS group was lower than in the control group (p < 0.001). The in-hospital mortality group also had a lower BSS score than the control group (p < 0.001) (Fig 2a and 2b).

When divided into quartiles based on BSS scores from low to high (Q1, Q2, Q3, Q4), unadjusted logistic regression analysis showed that compared to the low BSS Q1 group, the high BSS Q2 group had a reduced risk of ARDS [OR (95% CI) 0.616 (0.559–0.679), p < 0.001], the Q3 group had [OR (95% CI) 0.439 (0.393–0.490)], and the Q4 group had [OR (95% CI) 0.297 (0.269–0.328)]. As BSS increased, the risk of ARDS in the study population showed a downward trend. After adjusting for age, sex, laboratory indicators, and comorbidities, the high BSS Q4 group still had a significantly lower risk of ARDS compared to the low BSS Q1 reference group [OR (95% CI) 0.305 (0.276–0.337), p < 0.001] (Table 2).

In the overall elderly sepsis population, the study found that the BSS score was significantly negatively correlated with the risk of ARDS, showing that ARDS decreased as the BSS increased. The analysis revealed that for each unit increase in BSS, the risk of ARDS in elderly sepsis patients decreased by 15.7% [OR (95% CI) 0.843 (0.832–0.855), p < 0.001]. After adjusting for age, laboratory indicators, and comorbidities, the risk of ARDS continued to show a significant

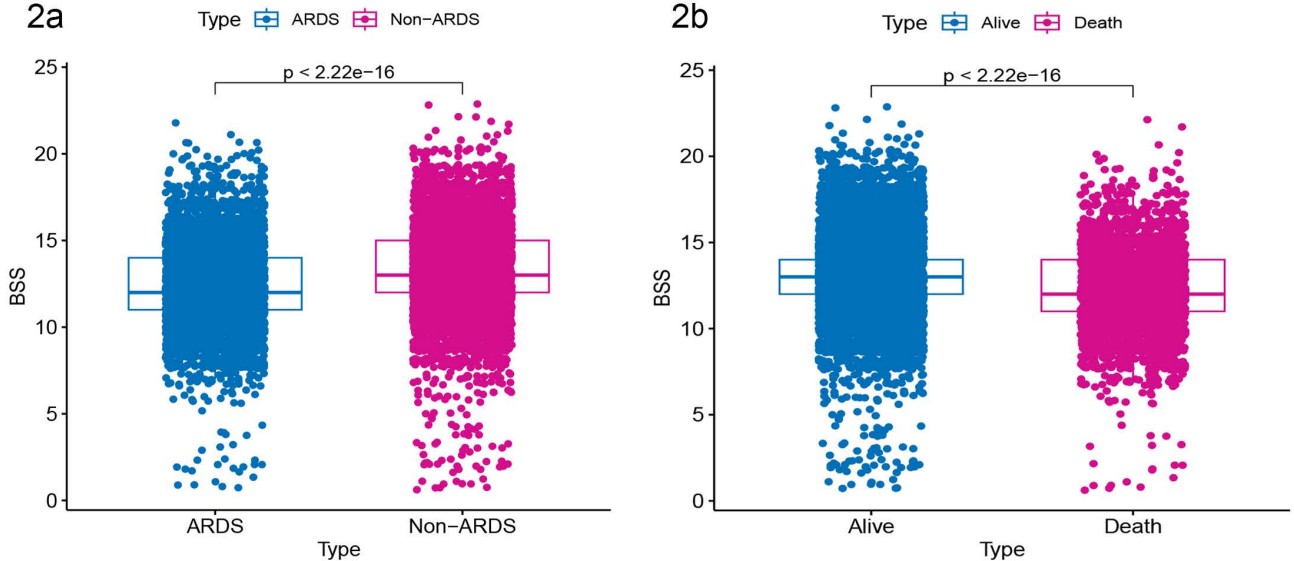

**Fig 2. Comparison of Braden Skin Scores (BSS) between patient groups. (a)** BSS scores were significantly lower in the ARDS group compared to the non-ARDS control group (p<0.001). **(b)** BSS scores were significantly lower in the in-hospital mortality group compared to the survival group (p<0.001).

**Table 2. Odds Ratios for associations between BSS and ARDS.**

|  | Q1 | Q2 | Q3 | Q4 |
|---|---|---|---|---|
| **Un-adjusted** | Ref. | 0.616(0.559-0.679) * | 0.439(0.393-0.490) * | 0.297(0.269-0.328) * |
| **Model 1** | Ref. | 0.618(0.560-0.681) * | 0.440(0.394-0.491) | 0.297(0.269-0.328) * |
| **Model 2** | Ref. | 0.632(0.573-0.697) * | 0.455(0.408-0.508) * | 0.308(0.278-0.340) * |
| **Model 3** | Ref. | 0.634(0.574-0.699) * | 0.453(0.405-0.506) * | 0.305(0.276-0.337) * |

Note. Model 1 was adjusted for age, sex.

Model 2 was adjusted for age, sex, BMI, WBC, Hemoglobin, Platelet, Albumin, Serum creatinine, ALT, AST.

Model 3 was adjusted for age, sex, BMI, WBC, hemoglobin, platelet, albumin, serum creatinine, ALT, AST, diabetes, hypertension, atrial fibrillation, COPD.

*P<0.001.

downward trend with increasing BSS [OR (95% CI) 0.845 (0.833–0.858), p<0.001]. Subgroup analysis by gender showed similar results in both male and female subgroups (Table 3).

### Association between BSS score and in-hospital

**Mortality risk in the ARDS subgroup.** In the overall elderly sepsis population, BSS score was significantly negatively correlated with in-hospital mortality risk, with mortality risk decreasing as BSS increased [OR (95% CI) 0.843 (0.830–0.857), p<0.001]. After adjusting for age, sex, laboratory indicators, and comorbidities, the in-hospital mortality risk continued to show a significant downward trend with increasing BSS [OR (95% CI) 0.849 (0.835–0.863), p<0.001]. Subgroup analysis in the ARDS population showed similar results in both ARDS and non-ARDS subgroups (Table 4).

**Sensitivity analysis.** To verify that our approach to handling missing data using imputation methods did not decisively influence the core conclusions, we re-analyzed the data, excluding individuals with any missing values across the analysis variables and retaining only cases with complete data for all variables. Using this dataset, we applied the

**Table 3. Odds Ratios for subgroup analyses between BSS and ARDS.**

| Subgroup | OR (95%CI) | P-value |
|---|---|---|
| **Total population** | | |
| Un- adjusted | 0.843(0.832-0.855) | <0.001 |
| Adjusted | 0.845 (0.833-0.858) | <0.001 |
| **Male** | | |
| Un- adjusted | 0.835 (0.820-0.851) | <0.001 |
| Adjusted | 0.836(0.820-0.852) | <0.001 |
| **Female** | | |
| Un- adjusted | 0.854(0.836-0.872) | <0.001 |
| Adjusted | 0.856 (0.838-0.875) | <0.001 |

Note. Model was adjusted for age, BMI, WBC, hemoglobin, platelet, albumin, serum creatinine, ALT, AST, diabetes, hypertension, atrial fibrillation, COPD.

**Table 4. Odds Ratios for subgroup analyses between BSS and mortality during hospitalization.**

| Subgroup | OR (95%CI) | P-value |
|---|---|---|
| **Total population** | | |
| Un- adjusted | 0.843 (0.830-0.857) | <0.001 |
| Adjusted | 0.849 (0.835-0.863) | <0.001 |
| **Non -ARDS** | | |
| Un- adjusted | 0.858 (0.838-0.880) | <0.001 |
| Adjusted | 0.859 (0.839-0.880) | <0.001 |
| **ARDS** | | |
| Un- adjusted | 0.886 (0.866-0.907) | <0.001 |
| Adjusted | 0.895 (0.875-0.917) | <0.001 |

Note. Model was adjusted for age, sex, BMI, WBC, hemoglobin, platelet, albumin, serum creatinine, ALT, AST, diabetes, hypertension, atrial fibrillation, COPD.

same multivariate regression model to evaluate the association between the BSS and the risk of ARDS. The results demonstrated that the complete-case analysis was highly consistent with the results from our primary analysis based on imputation.

## Discussion

This analysis of the MIMIC-IV database revealed a significant independent association between a lower Braden Skin Score (BSS) at ICU admission and an increased risk of both ARDS development and in-hospital mortality among elderly sepsis patients. The large cohort study (N = 16,565) demonstrated a substantial reduction in risks for both outcomes among patients with higher BSS scores, even after comprehensive adjustment for demographic and clinical covariates. These findings suggest that BSS may serve as a valuable early predictor of clinical outcomes in this vulnerable population.

Although a statistically significant age difference was observed between the ARDS and non-ARDS groups (74.0 vs. 74.6 years; p<0.001), the absolute difference of 0.6 years is not considered clinically meaningful, supporting the multifactorial nature of ARDS pathogenesis.

While BSS is traditionally used to assess pressure injury risk, its predictive value for pulmonary complications suggests shared underlying pathophysiological mechanisms. The association may be explained through several interconnected

pathways: First, sepsis is characterized by a dysregulated host response to infection, culminating in a cytokine storm. This systemic inflammation is a cornerstone of ARDS pathogenesis, damaging the alveolar-capillary membrane. Concurrently, pro-inflammatory cytokines (e.g., TNF-α, IL-1, IL-6) disrupt tissue repair mechanisms and increase skin vulnerability. Thus, a low BSS may reflect a state of heightened systemic inflammation that simultaneously predisposes the lungs to injury [21].Second, the Braden Scale components—mobility, sensory perception, and moisture—are sensitive to tissue perfusion. In sepsis, microcirculatory dysfunction and shock lead to inadequate tissue oxygenation. The skin, as a highly vascularized end-organ, is an early casualty, manifesting as reduced integrity (low BSS) [22]. Similarly, pulmonary micro-vascular thrombosis and endothelial injury are central to ARDS development. A low BSS may, therefore, serve as a visible marker of covert systemic microcirculatory failure that is also occurring in the lungs. Third, both the skin and the alveolar epithelium serve as critical barriers. Their integrity relies on adequate nutrition, perfusion, and cellular health. A low BSS indicates a failure to maintain cutaneous barrier integrity, often due to poor nutrition, immobility, and hypoperfusion—factors that also compromise the metabolic energy required for alveolar epithelial cell repair. The patient with fragile skin (low BSS) may have equally fragile pulmonary epithelium, more susceptible to insult in the setting of sepsis.

Furthermore, each unit increase in BSS score was associated with a 15.7% reduction in ARDS risk. These results indicate that BSS has substantial clinical utility as an indicator of ARDS risk in septic elderly patients, potentially serving as a sensitive assessment tool for evaluating overall physiological reserve and vulnerability to complications.

Additionally, our study reported a significant and negative correlation of the BSS with the risk of in-hospital mortality, which is consistent with the recent findings of Cheng et al. (2024) [23] and Shang et al. (2025) [24]. The results for the ARDS subgroup analysis were also accurate. This means that BSS is an indicator of the development of ARDS and is inversely related to patients' prognosis. The results for the ARDS subgroup analysis were also significant, meaning that BSS is an indicator not only of the development of ARDS but is also inversely related to patients' prognosis. As a result, evaluating BSS scores and their corresponding measures to prevent pressure injuries in elderly sepsis patients before them can help lower the risk of developing ARDS or in-hospital mortality.

Similar results were also observed in this study regarding the relationship between BSS score, ARDS risk, and in-hospital mortality risk in male and female subgroups. This only adds credibility to the BSS score as an indicator of prediction applicable across the board.

The findings of this study have important clinical implications. As a simple and practical tool, BSS could be used on ICU admission to rapidly identify elderly sepsis patients at high risk for ARDS and mortality, enabling earlier targeted interventions (e.g., optimized infection control, enhanced nutritional support, and close monitoring). Dynamic monitoring of BSS scores during the ICU stay could also provide insights into patient condition progression and treatment response.

Although this study provides evidence of a significant association between the BSS score and the risk of ARDS and in-hospital death, there are some limitations. First, this retrospective study includes patients' data only from the MIMIC IV database. Therefore, it has the potential for selection bias and limited generalizability of results. Second, although BSS is practical, raters may influence it differently, thus making it unsustainable. Third, other unconsidered potential influencing factors may also affect the observed associations despite adjusting for multiple confounding factors. These findings need future prospective, multicenter studies to validate and resolve these limitations.

Further research could explore whether BSS correlates with other clinical outcomes, including mechanical ventilation duration and ICU length of stay. Studies could explore using BSS with other biomarkers or risk prediction models to enhance accuracy and clinical utility. Future research could also examine the application value of BSS scores in different patient populations (i.e., patients outside of the elderly or septic population). This will support the broader use of BSS.

## Conclusion

This study concludes that elderly sepsis patients with a low BSS score on ICU admission have a significantly increased risk of ARDS and higher hospital mortality. As a simple and readily available assessment tool, a low BSS score is

associated with an increased risk of ARDS and mortality in elderly sepsis patients. This association suggests that BSS could potentially serve as an early warning indicator to help clinicians identify patients who may benefit from more vigilant monitoring and intensified supportive care. Therefore, we recommend that future prospective, multicenter studies validate the utility of BSS for early risk stratification in sepsis. If confirmed, interventional trials could investigate whether BSS-guided care protocols improve outcomes not only in septic patients but also in other critically ill populations.

## Supporting information

**S1 Table. Raw data study cohort.**
(XLSX)

**S1 Code. ARDS case extraction code.R.**
(R)

## Acknowledgments

The authors would like to express their gratitude to the Massachusetts Institute of Technology (MIT) and the Beth Israel Deaconess Medical Center for the creation, maintenance, and open access of the MIMIC-IV database. Without this invaluable resource, this study would not have been possible.

We are also deeply grateful to the editors and anonymous reviewers for their insightful comments and constructive suggestions, which have greatly improved the quality of this manuscript.

We acknowledge all the clinicians and researchers involved in the data collection for the MIMIC-IV database.

## Author contributions

**Data curation:** Yingqi Xiao, Jianwen Guo, Liufang Shu, Jingcheng Xu.

**Formal analysis:** Yingqi Xiao.

**Investigation:** Jianwen Guo.

**Methodology:** Yingqi Xiao.

**Supervision:** Zhiyong Li, Yongchun Li.

**Validation:** Yingqi Xiao, Jianwen Guo, Liufang Shu, Jingcheng Xu, Zhiyong Li, Yongchun Li.

**Visualization:** Zhiyong Li, Yongchun Li.

**Writing – original draft:** Yingqi Xiao, Yi Cao.

**Writing – review & editing:** Yingqi Xiao, Yi Cao.

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
