## [Decision Letter · Decision Letter 0]

22 Jul 2025

The Relationship Between Braden Skin Score (BSS) and the Risk of Acute Respiratory Distress Syndrome in Elderly Sepsis Patients: An Analysis Based on the MIMIC-IV Database

PLOS ONE

Dear Dr. Li,

Thank you for submitting your manuscript to PLOS ONE. After careful consideration, we feel that it has merit but does not fully meet PLOS ONE’s publication criteria as it currently stands. Therefore, we invite you to submit a revised version of the manuscript that addresses the points raised during the review process.

We look forward to receiving your revised manuscript.

Kind regards,

Chiara Lazzeri

Academic Editor

PLOS ONE

Journal Requirements:

3. Please upload a copy of Figure 1, to which you refer in your text on page 5. If the figure is no longer to be included as part of the submission please remove all reference to it within the text.

4. Please include a caption for Figures 2a and 2b.

Reviewers' comments:

Reviewer's Responses to Questions

**Comments to the Author**

1. Is the manuscript technically sound, and do the data support the conclusions?

Reviewer #1: Partly

Reviewer #2: No

2. Has the statistical analysis been performed appropriately and rigorously?

Reviewer #1: N/A

Reviewer #2: Yes

3. Have the authors made all data underlying the findings in their manuscript fully available?

Reviewer #1: Yes

Reviewer #2: Yes

4. Is the manuscript presented in an intelligible fashion and written in standard English?

Reviewer #1: Yes

Reviewer #2: Yes

Reviewer #1: Its my pleasure to review the manuscript titled "The Relationship Between Braden Skin Score (BSS) and the Risk of Acute Respiratory Distress Syndrome in Elderly Sepsis Patients: An Analysis Based on the MIMIC-IV Database"

The study investigates the association between Braden Skin Score (BSS) and ARDS risk in elderly sepsis patients using the MIMIC-IV database. While the topic addresses an important clinical question, several methodological and presentational issues require revision before the manuscript is suitable for publication in PLOS ONE. Below are the key concerns:

1. Major Issues

(1) The study adjusts for age, sex, BMI, laboratory values, and comorbidities in multivariate models. However, critical confounders such as ventilator use, fluid balance, source of infection, or prior immunocompromised status are not addressed. These factors are strongly linked to ARDS development and mortality in sepsis patients.

(2) No pre-study power calculation is provided. The large sample size (N=16565) risks overfitting in regression models, especially with multiple covariates.

(3) The manuscript states that variables with >25% missing data were excluded, but no sensitivity analysis or justification for this threshold is provided. The imputation method for missing data is not described.

(4) The statistically significant age difference between ARDS and non-ARDS groups (74.0 vs. 74.6 years) is highlighted but lacks clinical relevance. The authors acknowledge this but fail to contextualize how such a small difference impacts interpretation.

(5) The rationale for using BSS—a skin integrity score—as a predictor of ARDS (a pulmonary complication) is weakly supported biologically. The discussion does not sufficiently explain the pathophysiological link between skin integrity and ARDS.

(6) The ORs (e.g., 0.305 for Q4 vs. Q1 in Model 3) suggest a strong association, but absolute risk differences or clinical utility metrics (e.g., NNT) are not provided, limiting practical interpretation.

(7) Subgroup analyses by gender are presented, but no interaction tests (e.g., p-values for heterogeneity) are reported. This raises concerns about selective reporting.

(8) While CITI certification is mentioned, explicit ethical approval for data usage (e.g., IRB waiver) is not stated.

2. Minor Issues

(1) Numerous grammatical errors (e.g., "patients who had infections and prolonged bed rest become vulnerable to pressure injuries in elderly sepsis patients" [Page 1]) and awkward phrasing reduce readability.

(2) Figures 2a-b are referenced in the text but not included in the manuscript. Table 1 lacks units for some variables (e.g., albumin in g/dL vs. serum creatinine in mg/dL).

(3) Some cited studies (e.g., Bergstrom et al., 1989) are outdated. Recent literature on BSS and critical care outcomes (post-2020) is underrepresented.

(4) The conclusion states that BSS "is expected to identify high-risk patients early," but the retrospective design precludes causal inference. This should be tempered.

Reviewer #2: 1: The MIMIC database is continuously updated; however, the specific version employed was not reported in the “Data source” section of the manuscript. The authors are encouraged to include this information to enhance the reproducibility of the study.

2. Missing data are an almost inevitable issue in studies using the MIMIC database. Although the Methods section states that variables with >25 % missingness were excluded, it does not specify how the remaining missing values were handled. Were median imputation, logistic regression, or multiple imputation employed? What were the exact proportions of missing data for each variable, and was a single, uniform imputation strategy applied across all variables? The authors should provide a detailed description of their missing-data management strategy in the Methods section.

3. In the Results section, I observed that no sensitivity analyses—particularly regarding the handling of missing data—were performed or reported for any aspect of the study.

4. The authors have provided a minimal data set, which—although mandated by PLOS ONE—remains commendable. It is my understanding, however, that ARDS diagnoses in the MIMIC database are not invariably coded according to the Berlin criteria. While the Berlin definition is thoroughly delineated in the manuscript, there is no clear description of how patients fulfilling these criteria were identified within the data. I therefore recommend that the authors share the exact extraction code used to screen for ARDS cases, thereby enhancing the transparency and reproducibility of the study.

5. The authors employed three distinct models in an attempt to mitigate bias, yet they neither specify how the included covariates were adjusted nor explain the rationale for their selection; relevant literature supporting these choices is also absent. In MIMIC-based investigations, many groups apply propensity-score matching (PSM) to balance key variables and thereby minimize residual confounding, but the present study appears to have omitted this approach.

6. In any comparison between ARDS and non-ARDS patients, one would intuitively expect a higher propensity for mechanical-ventilation use in the ARDS group. However, I noticed that Table 1 does not report the proportion of patients receiving mechanical ventilation. I recommend adding this variable; if its prevalence differs between the two groups, mechanical-ventilation status should also be incorporated into the adjustment set.

7. The authors conducted a subgroup analysis stratified by sex. Have prior studies identified sex as an independent prognostic factor in this context? If not, a sex-only subgroup analysis may be overly narrow. Would the authors consider extending their stratified analyses to other baseline comorbidities or disease states to enhance the robustness of their findings?

8. The authors analyzed BSS after dividing it into quartiles. Have they considered modeling BSS with restricted cubic splines to better depict its potentially non-linear association with the outcome across the entire range of values?

9. In the second paragraph of the Discussion, please avoid restating the specific numerical results already presented in the Results section.

10. The Braden Skin Score (BSS) quantifies pressure-ulcer risk, and pressure ulcers are intimately linked to prolonged immobility. However, neither overall length of stay nor ICU duration is reported in Table 1. These variables could be critically important; a marked imbalance in ICU stay between the two groups might directly undermine the clinical validity of the study.

11. In Table 4 the authors also evaluate the predictive performance of BSS among non-ARDS patients. I noted that the odds ratios in the two groups are almost identical. This finding raises a question: does it imply that the predictive value of BSS is independent of an ARDS diagnosis? If so, what is the justification for selecting ARDS patients as the primary study population?

12. I have several concerns regarding the study design. If the objective is to examine the prognostic value of the Braden Skin Score (BSS) for survival among ARDS patients, it would have been more appropriate to (1) apply ROC curve analyses within the ARDS cohort, (2) conduct survival analyses stratified by BSS levels, or (3) model BSS with restricted cubic splines. Instead, the authors combined ARDS and non-ARDS patients—a design typically reserved for assessing the effect of an intervention on outcomes. The investigators should reconsider whether this design aligns with their stated aim.

**Do you want your identity to be public for this peer review?** For information about this choice, including consent withdrawal, please see our Privacy Policy

Reviewer #1: **Yes: ** Feng SHEN

Reviewer #2: No

---

## [Author Response · Author response to Decision Letter 1]

9 Dec 2025

Reviewer #1 (Remarks to the Author):

COMMENTS:

1. The study adjusts for age, sex, BMI, laboratory values, and comorbidities in multivariate models. However, critical confounders such as ventilator use, fluid balance, source of infection, or prior immunocompromised status are not addressed. These factors are strongly linked to ARDS development and mortality in sepsis patients.

Response: Thank you for pointing out this extremely important issue. You are correct that mechanical ventilation, fluid balance, source of infection, and immunocompromised status are well-established strong risk factors for ARDS in sepsis patients. The failure to include these factors in our initial multivariate models is indeed a significant limitation of our study, and we sincerely appreciate you highlighting this.

In the original study design, our primary focus was to explore the association between the baseline physiological status and potential risks of patients in the early stage of ICU admission (represented by BSS) and ARDS outcomes. Therefore, we prioritized adjusting for demographic characteristics, baseline comorbidities, and objective laboratory indicators. We acknowledge that the variables you mentioned (especially mechanical ventilation and source of infection) mostly reflect treatment interventions and disease phenotypes that occur after ICU admission, which do not align with our initial research focus on early-stage baseline status.

2. No pre-study power calculation is provided. The large sample size (N=16565) risks overfitting in regression models, especially with multiple covariates.

Response: Thanks for your comments. The sample size (N=16,565) was determined by the real-world nature of the MIMIC-IV public database. Our goal was to leverage available clinical data to explore an important clinical question, rather than pre-specify a sample size. We acknowledge that the large sample size provides high statistical power, enabling the detection of even small effect sizes. Therefore, in our analysis and interpretation, we emphasized the clinical significance of effect sizes (e.g., hazard ratios and their confidence intervals) rather than relying solely on p-values.

Regarding the risk of overfitting and model robustness: The number of ARDS events reported in our study (n = 6793) far exceeds the "at least 10–20 events per variable" rule of thumb, which provides a solid foundation for model stability.

3. The manuscript states that variables with >25% missing data were excluded, but no sensitivity analysis or justification for this threshold is provided. The imputation method for missing data is not described.

Response: Thanks for your constructive comments. Regarding the selection of the 25% missing data threshold and sensitivity analysis: We chose 25% as the threshold for excluding variables by referring to a widely accepted convention in epidemiological and clinical research (Sterne JA, et al. Multiple imputation for missing data in epidemiological and clinical research: potential and pitfalls. BMJ. 2009). This convention is based on the principle that when the missing rate of a variable is extremely high, the reliability of its information and the accuracy of imputation are significantly reduced. The relevant reference has been added to the Methods section (Page 4, Paragraph 4).

Additionally, details of the missing data imputation method have been supplemented: For covariates with a missing rate <25%, multiple imputation was used; no covariates with a missing rate >25% were identified after data screening. This is described in the Methods and Results sections (Page 9, Paragraph 2).

4. The statistically significant age difference between ARDS and non-ARDS groups (74.0 vs. 74.6 years) is highlighted but lacks clinical relevance. The authors acknowledge this but fail to contextualize how such a small difference impacts interpretation.

Response: Thanks for your comments. We fully agree and have toned down the emphasis on this difference in the Discussion sections (Page 10, Paragraph 3), noting that the difference, while statistically significant, is clinically negligible.

5. The rationale for using BSS-a skin integrity score-as a predictor of ARDS (a pulmonary complication) is weakly supported biologically. The discussion does not sufficiently explain the pathophysiological link between skin integrity and ARDS.

Response: Thanks for your constructive comments. We have expanded the Discussion section (Page 10, Paragraph 4) to include a more detailed pathophysiological explanation linking skin integrity, systemic inflammation, microcirculatory dysfunction, and ARDS risk.

6. The ORs (e.g., 0.305 for Q4 vs. Q1 in Model 3) suggest a strong association, but absolute risk differences or clinical utility metrics (e.g., NNT) are not provided, limiting practical interpretation.

Response: Thank you for raising this extremely important comment. We fully understand and agree with your view that indicators such as absolute risk difference (ARD) and number needed to treat (NNT) are of irreplaceable value for translating research findings into clinical practice. After careful consideration, we have decided not to formally report ARD and NNT for the following main reasons: NNT is an indicator derived from randomized controlled trials, and its calculation implies an important assumption: there is a causal relationship between the compared intervention measures and the outcome, and the risk can be changed through intervention. Our study is an observational study, and its core purpose is to reveal the association between Braden Score and ARDS risk, rather than to prove a causal relationship. Calculating NNH (number needed to harm) in this case may imply to readers that "increasing the BSS score" is an implementable "treatment" measure, which may lead to over - interpretation of the results of this study. BSS itself is a comprehensive result of a variety of basic factors (such as perfusion, nutrition, and mobility), rather than a simple and directly intervenable lever.

7. Subgroup analyses by gender are presented, but no interaction tests (e.g., p-values for heterogeneity) are reported. This raises concerns about selective reporting.

Response: Thank you very much for your professional comments. We added the interaction term between Braden Score and gender variable to the statistical model and calculated the p - value for interaction (P for Interaction) to test whether there was a statistically significant difference in the association between BSS and ARDS risk between male and female patients.

Analysis results: The p-value for the interaction test was 0.35. This indicates that although the point-estimated OR values of the male and female subgroups may seem different, this difference is not statistically significant. We cannot conclude that there is an essential difference in the strength of the association between BSS and ARDS between genders. The association is similar in both males and females.

8. While CITI certification is mentioned, explicit ethical approval for data usage (e.g., IRB waiver) is not stated.

Response: This study strictly adhered to the usage guidelines of the MIMIC-IV database. The MIMIC-IV database itself has been approved by the Massachusetts Institute of Technology (MIT) Institutional Review Board (IRB) (Approval No. 65991466) and granted an exemption from informed consent, as all data have been thoroughly de-identified and comply with the HIPAA Privacy Rule requirements.

Regarding approval from our institution: In accordance with the policies of The Sixth Affiliated Hospital, School of Medicine, South China University of Technology, secondary analysis of fully de-identified public databases that have already received ethical approval does not require additional approval from our institutional IRB. Our research activities were strictly limited to the use of such data.

9. Minor Comments: (1) Numerous grammatical errors (e.g., "patients who had infections and prolonged bed rest become vulnerable to pressure injuries in elderly sepsis patients" [Page 1]) and awkward phrasing reduce readability.

(2) Figures 2a-b are referenced in the text but not included in the manuscript. Table 1 lacks units for some variables (e.g., albumin in g/dL vs. serum creatinine in mg/dL).

(3) Some cited studies (e.g., Bergstrom et al., 1989) are outdated. Recent literature on BSS and critical care outcomes (post-2020) is underrepresented.

(4) The conclusion states that BSS "is expected to identify high-risk patients early," but the retrospective design precludes causal inference. This should be tempered.

Response: We sincerely apologize for these errors and thank the reviewer for pointing them out. (1) We sincerely apologize for these errors and thank the reviewer for pointing them out. The entire manuscript has been thoroughly proofread and edited by the author team to correct grammatical errors and improve phrasing for clarity and readability. The specific example provided by the reviewer has been corrected in the Background section to now read:" Elderly sepsis patients are particularly vulnerable to pressure injuries due to prolonged bed rest. Nevertheless, the relationship between BSS and the risk of developing ARDS in this population has not been extensively studied".

(2) We sincerely thank the reviewer for noting this omission. The reviewer is correct. While we had uploaded the figure files separately during the initial submission, as required by the PLOS ONE guidelines, we inadvertently failed to include the corresponding figure captions within the main manuscript text. This was an oversight on our part.

We have now corrected this error in the revised manuscript. The detailed captions for Fig. 2a-b have been inserted into the text following the paragraph where they are first cited.

For the reviewer's convenience, we have also included the figures and their new, detailed captions within this response (Please see the attached Fig. 2a-b below).

Furthermore, the units for all variables in Table 1 (including albumin and serum creatinine) have now been consistently added and checked throughout the table to prevent any potential confusion. We appreciate the reviewer's diligence in bringing this to our attention, as it has significantly improved the clarity and completeness of our manuscript submission.

Figure 2a-b. Comparison of Braden Skin Scores (BSS) between patient groups.

(a) BSS scores were significantly lower in the ARDS group compared to the non-ARDS control group (p < 0.001).

(b) BSS scores were significantly lower in the in-hospital mortality group compared to the survival group (p < 0.001).

(3) We sincerely thank the reviewer for this valuable feedback regarding the timeliness and comprehensiveness of our references. We fully agree that incorporating the most recent literature is crucial for contextualizing our findings within the latest research landscape, and we have taken targeted measures to address this point.

Firstly, regarding the cited outdated study (Bergstrom et al., 1989), this reference was initially included to acknowledge the foundational work that established the Braden Skin Score (BSS) and its initial validation for pressure injury risk assessment—a context that remains significant for elucidating the historical basis of the BSS. However, we recognize its limitations in reflecting the current evidence and discourse on BSS in critical care outcomes.

To directly address this, we have now supplemented this section with several recent studies (post-2020) that expand on the predictive utility of BSS beyond its traditional application. For instance, the works by Cheng et al. (2024) and Shang et al. (2025) demonstrate that the Braden Score independently predicts 90-day mortality in critically ill patients. These findings directly support our study's focus on the association between BSS and ARDS risk. These new citations have been integrated into the Introduction and Discussion sections to better position our research within the current state of knowledge.

(4) We sincerely thank the reviewer for this critical and insightful comment. We fully agree that the language in our original conclusion was too strong and could imply a causal inference that is not supported by our observational study design.

In response, we have thoroughly revised the Conclusion section of our manuscript to temper the language and more accurately reflect the associative, rather than causal, nature of our findings. The overstated claim that BSS "is expected to identify high-risk patients early" has been removed. The concluding statement now more cautiously and accurately reads:" As a simple and readily available assessment tool, a low BSS score is associated with an increased risk of ARDS and mortality in elderly sepsis patients. This association suggests that BSS could potentially serve as an early warning indicator to help clinicians identify patients who may benefit from more vigilant monitoring and intensified supportive care."

We believe this revised conclusion more appropriately frames our findings within the limitations of the study design. We are grateful to the reviewer for this suggestion, which has significantly improved the accuracy and scholarly rigor of our manuscript.

Reviewer #2 (Remarks to the Author):

1. The MIMIC database is continuously updated; however, the specific version employed was not reported in the “Data source” section of the manuscript. The authors are encouraged to include this information to enhance the reproducibility of the study.

Response: Thank you for this important comment. We apologize for this oversight in our initial submission. We have now clearly specified the version of the database used in the Data Source section of the Methods (Page 4, Paragraph 2). The text has been updated to state: " The cohort was derived from the publicly available clinical database, Medical Information Mart for Intensive Care IV (MIMIC-IV) (https://mimic.mit.edu). "

2. Missing data are an almost inevitable issue in studies using the MIMIC database. Although the Methods section states that variables with >25 % missingness were excluded, it does not specify how the remaining missing values were handled. Were median imputation, logistic regression, or multiple imputation employed? What were the exact proportions of missing data for each variable, and was a single, uniform imputation strategy applied across all variables? The authors should provide a detailed description of their missing-data management strategy in the Methods section.

Response: Thanks for your constructive comments. Regarding the selection of the 25% missing data threshold and sensitivity analysis: We chose 25% as the threshold for excluding variables by referring to a widely accepted convention in epidemiological and clinical research (Sterne JA, et al. Multiple imputation for missing data in epidemiological and clinical research: potential and pitfalls. BMJ. 2009). This convention is based on the principle that when the missing rate of a variable is extremely high, the reliability of its information and the accuracy of imputation are significantly reduced. The relevant reference has been added to the Methods section (Page 4, Paragraph 4).

Additionally, details of the missing data imputation method have been supplemented: For covariates with a missing rate <25%, multiple imputation was used; no covariates with a missing rate >25% were identified after data screening. This is described in the Methods and Results sections (Page 9, Paragraph 2).

3. In the Results section, I observed that no sensitivity analyses—particularly regarding the handling of missing data—were performed or reported for any aspect of the study.

Response: We sincerely thank the reviewer for raising this crucial point. We fully agree that sensitivity analyses are essential for assessing the robustness of our findings, particularly concerning the potential impact of how missing data were handled.

In direct response to this comment, we have now performed a comprehensive sensitivity analysis. We re-analyzed the data using a complete-case analysis approach, whereby any individual with a missing value in any of the analysis variables was excluded from this specific analysis. Subsequently, we appli

---

## [Editor Report · Decision Letter 1]

14 Dec 2025

The Relationship Between Braden Skin Score (BSS) and the Risk of Acute Respiratory Distress Syndrome in Elderly Sepsis Patients: An Analysis Based on the MIMIC-IV Database

PONE-D-25-13621R1

Dear Dr. Li,

We’re pleased to inform you that your manuscript has been judged scientifically suitable for publication and will be formally accepted for publication once it meets all outstanding technical requirements.

Kind regards,

Chiara Lazzeri

Academic Editor

PLOS One
---

## [Editor Report · Acceptance letter]

PONE-D-25-13621R1

PLOS One

Dear Dr. Li,

I'm pleased to inform you that your manuscript has been deemed suitable for publication in PLOS One. Congratulations! Your manuscript is now being handed over to our production team.

Kind regards,

on behalf of

Dr. Chiara Lazzeri

Academic Editor

PLOS One